# Comparative Studies of the Uptake and Internalization Pathways of Different Lipid Nano-Systems Intended for Brain Delivery

**DOI:** 10.3390/pharmaceutics15082082

**Published:** 2023-08-03

**Authors:** Ljubica Mihailova, Dushko Shalabalija, Andreas Zimmer, Nikola Geskovski, Petre Makreski, Marija Petrushevska, Maja Simonoska Crcarevska, Marija Glavas Dodov

**Affiliations:** 1Institute of Pharmaceutical Technology, Faculty of Pharmacy, Ss. Cyril and Methodius University in Skopje, Majka Tereza 47, 1000 Skopje, North Macedonia; lj.mihailova@ff.ukim.edu.mk (L.M.); d.shalabalija@ff.ukim.edu.mk (D.S.); ngeskovski@ff.ukim.edu.mk (N.G.); msimonoska@ff.ukim.edu.mk (M.S.C.); magl@ff.ukim.edu.mk (M.G.D.); 2Department of Pharmaceutical Technology and Biopharmacy, Institute of Pharmaceutical Sciences, University of Graz, Universitatplatz 1/EG, A-8010 Graz, Austria; 3Institute of Chemistry, Faculty of Natural Sciences and Mathematics, Ss. Cyril and Methodius University in Skopje, Arhimedova 5, 1000 Skopje, North Macedonia; petremak@pmf.ukim.mk; 4Institute of Pharmacology and Toxicology, Faculty of Medicine, Ss. Cyril and Methodius University in Skopje, 50 Divizija 6, 1000 Skopje, North Macedonia; marija.petrusevska@medf.ukim.edu.mk

**Keywords:** liposome, nanostructured lipid carrier, blood–brain barrier, neurons, internalization, cellular uptake, cytotoxicity

## Abstract

Lipid nano-systems were prepared and characterized in a series of well-established in vitro tests that could assess their interactions with the *hCMEC/D3* and *SH-SY5Y* cell lines as a model for the blood–brain barrier and neuronal function, accordingly. The prepared formulations of nanoliposomes and nanostructured lipid carriers were characterized by z-average diameters of ~120 nm and ~105 nm, respectively, following a unimodal particle size distribution (PDI < 0.3) and negative Z-potential (−24.30 mV to −31.20 mV). Stability studies implied that the nano-systems were stable in a physiologically relevant medium as well as human plasma, except nanoliposomes containing poloxamer on their surface, where there was an increase in particle size of ~26%. The presence of stealth polymer tends to decrease the amount of adsorbed proteins onto a particle’s surface, according to protein adsorption studies. Both formulations of nanoliposomes were characterized by a low cytotoxicity, while their cell viability was reduced when incubated with the highest concentration (100 μg/mL) of nanostructured lipid formulations, which could have been associated with the consumption of cellular energy, thus resulting in a reduction in metabolic active cells. The uptake of all the nano-systems in the *hCMEC/D3* and *SH-SY5Y* cell lines was successful, most likely following ATP-dependent internalization, as well as transport via passive diffusion.

## 1. Introduction

Much research has been conducted to examine and understand the anatomy and physiology of human brain cerebrovascular tissue, which is a remaining a major challenge in drug development for the treatment of brain diseases [1,2].

For the successful management of the central nervous system (CNS), diseases such as dementia, neurodegenerative diseases, epilepsy, panic attacks, meningitis, and brain tumors, high drug concentrations at the site of action have been of the utmost importance. Additionally, the pharmacological response provoked by potentially active substances depends on numerous factors, such as penetration through the blood–brain barrier (BBB) or their ability to attach onto specific protein transporters in order to facilitate their transfer across the membrane, uptake, and efficacy [3].

To achieve its pharmacological effect, the drug molecule needs to reach its target tissue in a sufficient quantity upon administration. However, many obstacles need to be overcome along the way, and the BBB is considered to be the most challenging obstacle for drugs acting on the CNS. The BBB is a specific and very dynamic system composed of endothelial cells accompanied by specialized tight junctions. It is not just a physical barrier that protects the brain from potentially harmful substances, but also poses an important gateway for nutrients and other molecules reaching the brain through various influx pathways or specific receptor-mediated trans or endocytosis [4,5].

It should be emphasized that many different drug molecules have shown pharmacological effects on the CNS, yet their delivery to the site of action is still debatable. In fact, a single modification in the drug molecule may lead to a significant change in the pharmacological response, as well as its behavior in vivo. A substantial polarity, high potential for hydrogen bond forming, molecular weight of >500 kDa, and the presence of rotating bonds or highly branched molecular structures have been pointed out as physical–chemical and molecular characteristics that lead to a reduced ability for penetration through the cells of the BBB and drug distribution in the CNS [6]. Moreover, recent research studies have shown that even a large percentage of newly discovered small molecules with potential effects on the CNS do not cross the BBB, thus leading to a lack of therapeutic concentrations in the brain and pharmacological response, respectively [7].

In the past, it was thought that the development of viable CNS treatment modalities should be based on the synthesis of highly lipophilic drugs with a small molecular weight, as they were considered to be the only molecules that could diffuse through the BBB. However, as the lipophilicity of molecules increases, their solubility and bioavailability reduce due to changes in the pharmacokinetic parameters of the drugs [8]. On the other hand, small water-soluble drugs cannot cross the BBB, as their penetration via the paracellular hydrophilic diffusion pathway usually terminates in the interendothelial space of the cerebrovascular tissue up to the tight junctions [9].

Recently, pharmaceutical technology has been more and more focused on introducing ways for the successful and safe delivery of conventional, already well-developed, and tested drugs to sites of action. Nanotechnology-based drug carriers are one of the non-invasive approaches that have been developing rapidly. There are many nano-systems with different drug release behavior that serve as carriers for efficient drug delivery [10].

The unique physical–chemical properties of nano-systems, including their size (~100 nm), flexibility, and large specific surface area, make them potential candidates for overcoming the barriers encountered by active substances directed to the brain. Due to their huge loading capacity, they have the ability to achieve a relatively high local concentration at a target site while reducing systemic side effects, which is particularly important for the treatment of brain diseases [11]. The wide palette of materials included in their composition has been increasing daily, from polymer and lipid nano-systems as already well-known carriers to nanostructures built from amphiphilic substances, metals, inorganic elements, carbon nanotubes, and dendrimers, etc. [12]. Moreover, they have a huge capability for passive or active BBB penetration, thus significantly improving the bioavailability and therapeutic effect of the drugs in much lower concentrations. Among them, lipophilic nano-systems, such as liposomes and nanostructured lipid carriers, stand out as one of the most attractive vehicles that can fuse with endothelial cells and transport active compounds via the transcellular pathway or endocytosis. In addition, the literature data suggest that lipid-based nano-systems may lead to reduced neuroinflammation, the activation of caspase-1 in microglial cells, and neurovascular damage [13]. As is well known, liposomes and nanostructured lipid carriers have been one of the most used and examined drug delivery systems over the years, due to their biocompatibility and biodegradability, the possibility of encapsulating a wide range of hydrophilic and hydrophobic substances, acceptable stability, and controlled and prolonged drug release [14,15].

In this direction, surface functionalization is one of the most effective ways to improve the pharmacokinetics and biodistribution of active compounds when they are loaded in lipid nano-systems. For instance, the addition of polyethylene glycol (PEG), poloxamer (POL), polyoxyethylene polymers, or polysaccharides, polyacrylamides, polyvinyl alcohol, and poly (N-vinyl-2-pyrrolidone) leads to the formation of a protective layer onto the nano-system’s surface, which further prevents their binding to plasma proteins, as well as reduces their uptake by the cells of the reticuloendothelial system, thus providing prolonged drug circulation time in vivo. In addition, recent scientific studies have shown that non-specific cell evasion could be achieved by the presence of these polymers, which further increases a drug’s bioavailability within the CNS parenchyma [16]. Hence, we thought that it would be interesting to conduct a comparative study and investigate the in vitro behavior of nanoliposomes and nanostructured lipid carriers, as well as their physical–chemical and biopharmaceutical characteristics when they have been coated with PEG or POL as polymers for steric stabilization.

Despite numerous studies conducted in recent years on lipid nano-systems and the materials used for their production, still, their safety cannot be fully guaranteed, taking into consideration all aspects from their physical–chemical and biopharmaceutical properties to all the biological characteristics of living organisms [17,18]. Moreover, in the literature, only separate studies on the efficacy of these systems could be found. 

Therefore, the main aim of this research was to compare the behavior of different lipid nano-systems characterized in terms of their physical–chemical properties and surface area, in a series of different, well-structured in vitro tests that could assess their interactions with the human cerebral microvessel endothelial cell line (hCMEC/D3) as a model for the BBB, and the human neuroblastoma clonal cell line (SH-SY5Y) as an in vitro model of neuronal function, thus evaluating their toxicity and potential for crossing the BBB and neuronal internalization.

## 2. Materials and Methods

### 2.1. Materials

Soybean lecithin (SL) was purchased from Vitalia, Macedonia. LIPOID PE 18:0/18:0-PEG 2000 (PEG) and hydrogenated soy phosphatidylcholine (LC-3) were kindly donated from Lipoid, Germany. Phospholipone 90H (PL90H) was supplied from Phospholipid, Germany, Tween 80 from Merck, Germany, and Poloxamer 407 (POL) from BASF, Germany. Cholesterol (CH), oleic acid (OA), bovine serum albumin (BSA), Bradford Reagent, Dulbecco’s Phosphate Buffered Saline (DPBS), collagen type I from rat tail, fluorescein isothiocyanate isomer I (FITC), Nile Red, chlorpromazine, and indomethacin were obtained from Sigma Aldrich (St. Louis, MO, USA). The immortalized human cerebral microvascular endothelial (hCMEC/D3) cell line (CELLutions, Biosistems/Cedarlane^®^, Burlington, ON, Canada) was maintained in Endothelial Basal Medium-2 (EBM-2), supplemented with Fetal Bovine Serum (FBS), chemically defined lipid concentrate, HEPES 1M, penicillin–streptomycin (Life Technologies, Carlsbad, CA, USA), human basic Fibroblast Growth Factor (bFGF), ascorbic acid, and hydrocortisone (Sigma Aldrich, St. Louis, MO, USA). Trypsin-EDTA was purchased from GIBCO, Thermo Fisher scientific (Waltham, MA, USA). The human neuroblastoma clonal cell line (LCG Standards, Wesel, Germany) was grown in Dulbecco’s Modified Eagle Medium (DMEM), supplemented with 10% FBS, 1% penicillin–streptomycin (Life Technologies), and 0.2% non-essential amino acids (NEAA) (Sigma Aldrich, USA). CellTiter 96 AQueous Non-Radioactive Cell Proliferation Assay (MTS) and CytoTox-ONETM Homogeneous Membrane Integrity Assay (CytoTox) were provided by Promega, Madison, WI, USA. Alexa Fluor^TM^ Phalloidin 488, Hoechst Fluorescent stain, Dil stain (1,1′-Dioctadecyl-3,3,3′,3′-Tetramethylindocarbocyanine Perchlorate (‘DiI’; DiIC18(3))), pHrodo^TM^ Green dextran conjugate, and CellLightTM ER-GFP were purchased from Thermo Fisher Scientific, Waltham, MA, USA. Human plasma from healthy volunteers was obtained from the Institute for Neurology, Clinical Center—Mother Theresa, Skopje, R. N. Macedonia. 

All the other chemicals and reagents were of the highest purity grade commercially available and used as received.

### 2.2. Preparation of Nanoliposomes (NL)

Two formulations of NL were prepared using the modified lipid film hydration technique (Appendix A) [19]. In the first step, SL, PC-3, CH, and PEG (NLPEG) and SL, PC-3, CH, and POL (NLPOL), in mass ratio of 17.3:1:1:2, accordingly, were dissolved in chloroform/methanol mixture in a ratio of 3:2 (*v*/*v*). Additionally, 7.5 mg of DIL was added in the organic phase for the purpose of the internalization studies. Afterward, the organic solvents were evaporated under vacuum using a rotavapor (40 °C, 50 rpm; Buchi 215, Flawil, Switzerland). The obtained dried lipid film was hydrated with 1 M phosphate buffer at pH 7.4 (Ph. Eur. 11.0) in three consecutive cycles (10 min), under two hydration steps: ultrasonication (50/60 Hz) at 40 °C for 5 min with gentle shaking (5 min). The obtained vesicles were homogenized (24,000 rpm, 3 min; Ultra-Turrax T25, Ika-Werke, Germany) and allowed to stand at 2–8 °C for 24 h. After 24 h, the samples underwent homogenization with ultra-turrax again (3 min on 6500 rpm) and were stored at 4–8 °C in a refrigerator.

### 2.3. Preparation of Nanostructured Lipid Carriers (NLC)

The NLC formulations were prepared using the solvent evaporation method (Appendix A) [19]. The oil phase, composed of PL90H and OA (1:1.15 mass ratio, respectively), was dispersed in methanol under magnetic stirring (250 rpm, 50 °C; Jenway, Bridgnorth, UK). Additionally, 4 mg of DIL was added in the oil phase for the purpose of the internalization studies. Subsequently, the oil phase was added dropwise to the water phase, which consisted of 3% Tween 80 water solution and 35 mg PEG in the first formulation (NLCPEG), and 3% Tween 80 water solution and 35 mg POL in the second formulation (NLCPOL), under continuous magnetic stirring at 50 °C and 200 rpm for a period of 2 h. The obtained lipid emulsions were homogenized with ultra-turrax for 5 min at 6500 rpm and allowed to stand for 24 h in a refrigerator at 4–8 °C for the recrystallization of the lipids and the formation of the final particle structure.

### 2.4. Morphological Properties of Prepared Nano-Carriers

The morphological properties and surface appearance of the prepared samples were examined using transmission emission microscopy (TEM) (JEM-1400, Jeol, Japan) attached to a digital camera (Veleta TEM Camera, Olympus, Hamburg, Germany) and controlled by iTem software v.5.2. Briefly, a small quantity of the dispersion (2 µL) was placed on a coated copper grid (100 mesh) with an additional carbon film.

### 2.5. Particle Size, Particle Size Distribution, and Surface Potential of the Prepared Nano-Carriers

The mean particle size (z-average diameter—nm), particle size distribution (polydispersity index—PDI), and zeta potential (mV) of the nano-carriers were determined using Zetasizer Nano Series (Nano-ZS, Malvern Instruments Ltd., Malvern, UK). In this sense, each sample was diluted 20 times in phosphate buffer at pH 7.4 (Ph. Eur. 11.0) with a low molarity (10 mM). In order to investigate the influence of the polymer for steric stabilization on the in vitro stability of the prepared NL and NLC and their protein corona formation, the prepared samples (200 μL) were incubated in 800 μL of 1 M phosphate buffer at pH 7.4 (Ph. Eur. 11.0) or human plasma (100%) at 37 °C for 1, 4, 6, and 24 h. After incubation, the mean particle size and particle size distribution of the samples were measured. The measurements were performed in disposable transparent cuvettes at 25 °C with a thermostating time of 120 s, viscosity of the medium of 0.8894 cP, dielectric constant of 78.5, and angle of 173°. At least 3 batches from each sample and at least 3 analyses from each batch were measured. Each analysis had an average value of at least 12 measurements.

### 2.6. ATR-FTIR Spectroscopy Analysis

In order to gain insight into the structures of the prepared formulations, as well as examine the interactions between the components, an ATR-FTIR analysis was performed. The NL and NLC, as well as each component alone, were firstly stored at −80 °C for 2 h, were then subsequently submitted to freeze-drying (−40 °C, 0.052 mBar; FreeZone 2.5 Freeze Dry System—LABCONCO, Kansas City, MI, USA). The ATR-FTIR spectra were recorded on a Perkin-Elmer System 2000 FT-IR spectrometer, using Golden Gate (Specac) as an ATR accessory, equipped with a diamond ATR crystal and ZnSe lenses, with 45° angle of incidence. The number of scans was 64, both for the background and the sample spectra.

### 2.7. Protein Adsorption Studies

Experiments were performed to investigate the steric stabilization effect of the surface-oriented polymer (PEG and POL, respectively) on the protein corona formation, as well as to predict the possible stability and in vivo fate of the nano-systems. The experimental procedure was performed in accordance with previously established experimental conditions [20,21]. In this context, the amount of adsorbed proteins on the prepared NL and NLC surfaces was compared to their analogs without a polymer (NL0 and NLC0). The formation of the bonds between the nano-systems and plasma proteins was determined by incubating the prepared samples with human plasma from healthy volunteers. The NL and NLC formulations (a final lipid concentration of 1 mM) were incubated with plasma (1:1 (*v*/*v*)) for 1 h at 37 °C on a water bath with constant horizontal shaking (75 rpm). Before the incubation step, the human plasma was stored at 4 °C and then centrifuged for 5 min at 15,000 rpm, so the protein aggregates could be removed. The absorbed amount of protein/mg lipid was calculated indirectly by determining the non-bound amount of protein in the filtrate after centrifugation (15 min, 4000 rpm, and 3 washing cycles) using the Bradford method (microtiter plate protocol) (VICTOR Perkin Elmer, Shelton, CT, USA) and consequent subtraction from the initial protein amount in the plasma. Diluted plasma, incubated under previously mentioned conditions, was used as a control. The samples and controls were applied in triplicate. The percent of adsorbed proteins was calculated by the division of the amount of adsorbed proteins onto the NL and NLC surfaces with the previously determined total amount of proteins in the plasma, multiplied by 100.

### 2.8. In Vitro Cell Culture Studies

#### 2.8.1. hCMEC/D3 Cell Culture Lines

In vitro uptake and permeability studies were performed on an hCMEC/D3 cell culture. The cells were cultured in a supplemented EBM-2 (37 °C, atmosphere of 5% CO_2_) in T-75 cell culture flasks (Greiner Bio-One GmbH, Germany), pre-coated with 0.05 mg/mL of collagen type I from rat tail in DPBS. The cell medium was replaced every 2–3 days until the cells became confluent. The cells were detached from the culture flask walls after incubation (37 °C for 8 min) in 0.1 mg/mL of a trypsin–EDTA solution. The cell suspension was centrifuged at 1500 rpm for 3 min and the supernatant was removed. Subsequently, the cells were suspended in 5 mL of medium and were ready for further experiments.

#### 2.8.2. SH-SY5Y Cell Culture Lines

The human neuroblastoma cell line (SH-SY5Y) was cultured in DMEM at 37 °C in an atmosphere of 5% CO_2_. The cell medium was replaced every 2–3 days until reaching confluence. During the splitting step, the media were removed, and the cells were detached from the walls of the culture flask after incubation (37 °C for 3 min) in a 0.1 mg/mL trypsin–EDTA solution. The obtained cell suspension underwent centrifugation at 800 rpm for 5 min and the supernatant was removed. Consequently, the cells were suspended in 5 mL of medium and were ready for further experiments.

#### 2.8.3. Cell Viability Assay (MTS Assay)

The potential cytotoxic effect of the lipid nano-systems on the hCMEC/D3 cells was observed using an MTS assay. The cells were seeded in a 96-well plate (1 × 10^4^ cells) pre-coated with Collagen Type I, Rat Tail, and grown at 37 °C in an atmosphere of 5% CO_2_ in EBM-2 (200 μL). After reaching approximately 70–80% confluence, the cells were incubated with different concentrations of the prepared lipid nano-systems for 4, 24, and 48 h. Upon the expiration of each time interval, the medium of each well was separated from the cells and stored for a lactate dehydrogenase (LDH) assay, and the cells in the wells were treated with 20 μL of an MTS solution for 4 h (37 °C and 5% CO_2_). The absorbance was measured using a microplate reader (BMG Labtech, Ortenberg, Germany) at a wavelength of 490 nm. The cell viability was expressed as a % compared to the cells incubated only with EBM-2 (negative control). Triton X-100 was used in the MTS assay as the positive control, since its detergent disrupts the cell’s membrane.

#### 2.8.4. Cell Cytotoxicity Assay (LDH Assay)

To examine whether the cytotoxic effect was due to cell necrosis, an LDH assay was performed. The cell culture medium (25 μL) was withdrawn after the incubation of the NL and NLC and was transferred into white 96-well plates. The release of LDH into the culture supernatants was detected by adding 25 μL of substrate (cytotox reagent—blue). After 20 min of incubation at 37 °C in an atmosphere of 5% CO_2_, 13 μL of stop solution was added to each well. The fluorescence was measured at 560 nm excitation and 590 nm emission. The cytotoxicity was expressed as a % compared to the maximum LDH release in the presence of Triton X-100 (positive control). EBM-2 was used as a negative control, since no cytotoxicity was detected in such conditions.

#### 2.8.5. Life-Cell Imaging (Fluorescent Microscopy)

In order to investigate the internalization of the prepared lipid nano-systems in living cells, SH-SY5Y cells were previously seeded (2 × 10^5^ cells per well) on 35 mm glass dishes (μ-Dish 35 mm, Polymer coverslip uncoated) and incubated at 37 °C and 5% CO_2_ for 48 h, with green fluorescence for visualization of the endoplasmic reticulum (CellLight ER-GFP, BacMam 2.0, Thermo Fisher). Afterwards, the cells were incubated for 1 h with the NL and NLC prelabeled with Dil red dye (1,1′-Dioctadecyl-3,3,3′,3′-Tetramethylindocarbocyanine Perchlorate).

In order to examine the internalization and colocalization of the particles, the cells were pre-treated with pHrodo Green dextran conjugate, which gives green fluorescence in an acidic environment, i.e., a color characteristic for visualization of the endocytic pathway. The incubation of the cells with the Dil-labeled NL and NLC lasted for 4 h. 

After the appropriate incubations, the cells were rinsed twice with phosphate buffer at pH 7.4 and images were taken using fluorescence microscopy at 37 °C on a Zeiss Axio Observer Z1 inverted microscope (Zeiss, Jena, Germany), equipped with an epifluorescence illuminator and chamber heating plate. The images were further processed using Carl Zeiss software (ZEN 2.6).

#### 2.8.6. Confocal Microscopy

In order to visualize the internalization and depict the intracellular localization of the prepared formulations in BBB cells and neurons, the cellular uptake of the NL and NLC by the hCMEC/D3 and SH-SY5Y cell lines was investigated using Confocal laser scanning Microscopy imaging (Carl Zeiss, Axiovert 200 M Inverted Microscope). The cells were grown on 35 mm glass-bottom dishes (2 × 10^5^ cells per well) and incubated for 4 h with 10 μg/mL of the prepared NL and NLC previously labeled with Dil. After incubation, the cells were washed twice with PBS and fixed with 3.7% paraformaldehyde for 20 min at room temperature, followed by two washing steps with phosphate buffer at pH 7.4. Afterward, the cell membranes of both the cell culture lines were stained with Alexa Fluor^TM^ Phalloidin 488 for 10 min at 37 °C, followed by staining of the nuclei with Hoechst fluorescent stain (hCMEC/D3) and 300 nM DAPI (4′,6-diamidino-2-phenylindole) (SH-SY5Y) for 5 min at room temperature. Vectashield was used as a mounting medium. The images were processed using Carl Zeiss software (ZEN 2.6). Alexa Fluor 488 Phalloidin was used to stain the actin cytoskeleton and was excited at 488 nm and detected using a bandpass filter (BP 505/550 nm) for the green channel. Hoechst Fluorescent stain and DAPI, respectively, were used to counterstain the nucleus and were excited at 405 nm and detected using a bandpass filter (BP 420/480 nm) for the blue channel. The Dil-labelled samples were detected at a 549 nm excitation wavelength using a longpass filter (LP 560 nm) for the red channel.

#### 2.8.7. Quantitative Cell Uptake Studies

For the aim of the in vitro cell uptake studies of the NLs and NLCs, hCMEC/D3 and SH-SY5Y were seeded on 96-well plates (10^4^ cells/well) and cultured in 0.2 mL of EBM-2 and DMEM, accordingly. After reaching confluence (2 days incubation at 37 °C in an atmosphere of 5% CO_2_), the cells were incubated with previously Nile-Red-labeled NLs and NLCs diluted in a suitable cell culture medium (10 μg/mL) for 2 h at 37 °C in an atmosphere of 5% CO_2_. Afterwards, the cells were washed twice with PBS and lysed with 2% Triton X-100 (2 h, 37 °C, and 5% CO_2_). The fluorescence was measured at 535 nm excitation and 635 nm emission (BMG Labtech, Ortenberg, Germany). Cells incubated only in the medium were used as a blank. The amount of internalized nano-formulations was calculated as a % from the maximum fluorescence obtained using the same concentration of the suitable, non-incubated, native NL and NLC formulations. 

In order to determine the endocytic uptake mechanisms, these experiments were also carried out in the presence of endocytic inhibitors. Prior to the incubation step, one group of cells was pretreated with 15 μM of chlorpromazine for 40 min, followed by incubation with NLs and NLCs, while another group of cells was pretreated with 25 μM of indomethacin for 40 min (37 °C, 5% CO_2_). For observing the effect of low temperature, a general metabolic inhibitor was used and one group of cells was maintained at 4 °C prior (40 min) and during (2 h) the incubation with the nano-formulations.

#### 2.8.8. Statistical Analysis

Partial least square (PLS) methodology was applied using the validated statistical software Simca 14.1 (Sartorius Stedim Biotech, Goettingen, Germany). VIP scores were used with the aim of pointing out the dominant factors that had significant contributions to the model. The differences among the groups were evaluated using a *t*-test or one-way analysis of variance (ANOVA), where *p* < 0.05 was considered to be significant.

## 3. Results and Discussion

### 3.1. Morphological Properties

The shape of the nano-carriers was revealed to be one of the parameters that strongly influenced their biological behavior, besides their size and surface charge [22]. In this context, it was of great importance for the prepared nano-delivery systems to be extensively characterized before their use to correlate their physical–chemical properties with their in vitro performance. TEM photomicrographs revealed that all four nano-formulations were characterized by spherical and symmetric shapes (Figure 1).

### 3.2. Particle Size, Particle Size Distribution, and Surface Potential

The NLs samples prepared using the modified lipid film hydration technique had a z-average diameter of ~120 nm and followed a narrow unimodal distribution with a PDI value ranging from 0.235 ± 0.035 to 0.270 ± 0.006 for NLPEG and NLPOL, accordingly (Appendix A). On the other hand, the NLC formulations resulted in lower z-average diameter of ~105 nm with a PDI of ~0.190 (Table 1 and Appendix A). The obtained results are favorable, since the literature data suggest that spherical nanoparticles with a particle size of <200 nm are characterized by a reduced uptake by the RES system and a prolonged circulation time in human plasma [23], which results in a greater possibility for the interaction of the nanoparticles with the specific structures on the luminal side of the BBB endothelial cells, and consequently, successful transport across the BBB. Moreover, it has been reported that nano-systems with a particle size diameter ranging from 100 nm to 200 nm have maybe been the most suitable as drug carriers for achieving the highest cellular uptake [24].

The ZP was negative for all the formulations, ranging from −24.30 ± 0.850 to −31.2 ± 3.326 mV for NLCPOL and NLPOL, accordingly (Table 1 and Appendix A), which could be attributed to the negatively charged groups of the structural lipids of the formulations [25]. According to the findings of Patil et al. [26], negatively charged nanoparticles show a lower protein adsorption onto surfaces, implying a higher chance for achieving successful BBB delivery.

Upon the intravenous administration of nanoparticles in vivo, the adherence of serum biomolecules on the nanoparticles’ surfaces could result in particle integrity loss and could severely alter the particle properties, which has a major influence on their key performances, including their aggregation behavior, circulation time in vivo, toxicity, and targeting capability [27]. 

Therefore, with the aim of obtaining formulations that would be characterized by controlled drug release and achieve efficient accumulation in the brain by passive targeting in vivo, it was of high importance to explore the in vitro stability of the NL and NLC, not only in the formulation vehicle (physiologically relevant medium at pH 7.4), but also in the human plasma. Taking into consideration that a determination of mean particle size is an effective tool for predicting and observing the agglomeration process, the long-term stability of the nano-formulations, z-average diameter (nm), and PDI of the NL and NLC formulations were measured at the beginning of the experiment and at different time intervals (2, 4, 6, and 24 h) after their incubation in the abovementioned media.

From the obtained results, it could be concluded that all the formulations showed an appropriate stability in a physiological relevant medium with a pH of 7.4 (Ph. Eur. 11.0), except for NLCPOL, where there was noted a significant change in the z-average diameter after 24 h of incubation time (*p* = 0.0101; *p* < 0.05; ANOVA) (Figure 2a). Similar values for the nano-systems’ average diameters were obtained when they were incubated in human plasma, except with NLCPOL, where, after 24 h, there was an increase in particle size of ~25%, reaching 157.1 nm (*p* = 0.0000; *p* < 0.05; ANOVA) (Figure 2b). This significant increase in particle size after the incubation of the samples in human plasma was probably related to the formation of protein corona, thus leading to physical instability of the system [28]. 

From the presented results, it can be concluded that the prepared lipid nano-systems showed an appropriate stability with a 24 h incubation time in phosphate buffer at pH 7.4, as well as in the human plasma, except for the NLC sterically stabilized with POL. These results indicate the possible influence of the polymer for steric stabilization on the stability of the nano-systems. This phenomenon could be explained by the differences in the chemical structure and spatial arrangement of the two polymers within the lipid nano-carriers. Namely, in the samples covered with PEG2000-DSPE, PEG groups are anchored to a phospholipid tail (DSPE), which is aligned parallel within the membrane phospholipid molecules of the formulations, allowing for the hydrophilic PEG chains to extend outward. This configuration was shown to decrease the surface hydrophobicity, and therefore reduce the plasma protein adsorption [21]. On the other hand, the large hydrophobic polypropylene oxide block of Poloxamer was probably positioned deeper and closer to the lipid layer of the NLs and NLCs, thereby limiting the degree to which the PEG chains were projected and extended on the surface of the nano-formulations [29].

According to the literature data, upon the intravenous administration of nanoparticles in vivo, the adherence of serum biomolecules on the nanoparticles’ surfaces could result in particle integrity loss and severely alter the particle properties, which has a major influence on their key performances, including their aggregation behavior, circulation time in vivo, toxicity, and targeting capability [27]. In this sense, the change in the particle diameter during the incubation in human plasma may have been due to the destabilization of the system in terms of particle agglomeration, or, more likely, the formation of a protein corona [28]. This phenomenon could be observed at the first time points of the incubation of the formulations, which points to the dynamic process of protein corona formation, i.e., the constant adsorption and desorption of proteins from the particles’ surfaces, which is especially emphasized during the formation of the initial, “soft” corona [30].

### 3.3. ATR-FTIR Spectroscopy Analysis

The spectra of the prepared formulations, as well as the spectra of the individual components composing them, were recorded. As can be seen in Figure 3a, the main contribution in the NL samples came from soybean lecithin (the predominant component in the formulation), hydrogenated soybean phosphatidylcholine, cholesterol, and the stealth polymer DSPE-mPEG-2000 (NLPEG) or poloxamer 407 (NLPOL). In the region between 2960 and 2850 cm^−1^, symmetric CH_2_ stretching bands were present in all the components, as well as in the spectra of the prepared NLs. The presence of DSPE-mPEG 2000 was confirmed by the appearance of a band at 961 cm^−1^, attributed to the aliphatic P-O-C stretching vibrations, which is minorly shifted in the spectrum of the NLPEG formulation (970 cm^−1^). Furthermore, the band at 1100 cm^−1^, ascribed to the C-O secondary alcohol vibration in the starting component, appeared in the NLPEG spectrum with a lower intensity and was slightly displaced (1092 cm^−1^), probably due to the involvement of the corresponding oxygen atom with an interaction with the other components that weakened the strength of the C-O bond. Moreover, the C=O vibrations of the DSPE-mPEG 2000 also appeared at 1738 cm^−1^ in the spectra obtained from the NLPEG formulation. The presence of poloxamer 407 in the NLPOL formulation was confirmed by the band at 841 cm^−1^ and was assigned to the PEO chain of the poloxamer [31]. This spectral window in the formulation was not interfered with by any other band of the other excipients [32].

The ATR-FTIR spectra of the NLCPEG and NLCPOL (Figure 3b) show the presence of bands of the oleic acid and phospholipon, both characterized by nearly identical spectral features in the 3000–2800 cm^−1^ region. The predominant absorption here arose from the sharp bands at about 2916 and 2850 cm^−1^, originating from the asymmetric and symmetric CH_2_ stretching vibrations, respectively. Moreover, the spectra of the two aforementioned components differed in the position of the C=O carbonyl stretching vibration at 1727 cm^−1^ (ester group) in the phospholipon and 1710 cm^−1^ (carboxyl group) in the oleic acid. However, broader bands in this region are seen in the spectra of the NLC formulations (1738 cm^−1^ and 1735 cm^−1^ for NLCPEG and NLCPOL, respectively), which could be explained by the overlap of the discussed carboxyl stretching bands [32]. The presence of DSPE-mPEG 2000 in the NLCPEG and poloxamer 407 in the NLCPOL could be confirmed by the weak bands at 1115 cm^−1^ and 1061 cm^−1^ in the NLCPEG and 1144 cm^−1^ in the NLCPOL (upper band from the C–O and lower band from the C–C stretching vibrations). It should be mentioned that these bands were characterized by a lower intensity compared to the spectra of the pure stealth polymers, suggesting possible conformation changes in the polymers upon their incorporation into the nano-formulations.

### 3.4. Protein Adsorption Studies

In the past few decades, it has been well shown that the protein corona formation on nanoparticles’ surfaces influences their in vivo fate and may significantly interfere with the targeting yields, biodistribution, and cell internalization of nano-delivery systems [33,34]. Additionally, many research studies have confirmed that PC formation and evolution generally depend on nanoparticle properties, such as: size, surface charge, morphology, hydrophobicity, and the presence of ligands [35]. Therefore, it was important to obtain relevant information regarding the interactions between the prepared nano-carriers and human plasma proteins, in order to predict the process of PC formation in vivo. For this purpose, samples containing PEG (NLPEG and NLCPEG), nano-carriers covered with POL (NLPOL and NLCPOL), and formulations with no PEG or POL in the content (NL0 and NLC0) were prepared and incubated for 1 h in human plasma from healthy volunteers. The results from the protein adsorption studies are given in Table 2.

From the results presented in Table 2, it can be concluded that the formulations prepared with no stealth polymer on their surface tended to have a significantly higher % of plasma protein adsorption (*p* = 0.0031 and *p* = 0.0290 for NL and NLC, respectively; *p* < 0.05; ANOVA) compared to the formulations containing PEG (NLPEG and NLCPEG) or POL (NLPOL and NLCPOL). The obtained results were expected, since PEGylation has been a traditionally used method for reducing the interaction of nanoparticles with plasma proteins [21,36]. Similar to PEG, the hydrophilic blocks of POL on the hydrophobic surfaces of lipid nanoparticles are capable of forming a hydrophilic barrier for protein adsorption, resulting in reduced mononuclear phagocyte system (MPS) uptake and thus achieving a prolonged blood circulation time [20,37,38,39].

On the other hand, it was revealed that there were no statistically significant differences between the percentage of the adsorbed plasma proteins onto the formulations covered with PEG (*p* = 0.1448; *p* < 0.05; *t*-test) and POL (*p* = 0.3950; *p* < 0.05; *t*-test) used as stabilizing polymers.

### 3.5. Cell Viability Assay (MTS Assay)

The continuous application of lipid nanoparticles in the treatment of different diseases must be accompanied by basic studies on any potential toxicity risks in order to determine the optimal, safe concentrations for use. It is well known that cellular uptake depends on the particles’ lipid composition and chemical structure, as well as their size, which further raises the interest into their associated cellular toxicity [40]. 

The results obtained from the MTS reduction assay used to assess the cell viability after 4, 24, and 48 h of exposure at three different concentrations of the prepared lipid nano-systems are presented in Figure 4. It can be seen that, after treating the cells with the highest concentration of NLCPEG and NLCPOL (100 μg/mL) at all the time points examined, the cell viability decreased to 26.05 ± 11.11% and 25.34 ± 1.12% at 4 h, 12.18 ± 3.03% and 10.96 ± 1.04% at 24 h, and at 48 h, 7.47 ± 1.23% and 7.83 ± 0.49%, for NLCPEG and NLCPOL, respectively. A significant decrease in the cell viability (31.16 ± 7.10%) was also observed after 48 h of incubation with the highest concentration of NLPEG (100 μg/mL) (Figure 4c).

Based on the obtained results, it can be concluded that the nanostructured lipid carriers (NLCPEG and NLCPOL) tended to reduce the cell viability at a higher % than the liposomal vesicles (NLPEG and NLPOL). The obtained results are in accordance with previously published work claiming the decisive role of Tween 80 (the main component in NLCPEG and NLCPOL) in the overall toxicity of nanoparticles [41,42]. 

Regarding the results for the viability of the MTS test, the multivariate analysis with PLS showed that the concentration of the sample was the dominant factor that influenced the results themselves (Appendix A). None of the other experimental conditions, nor the type of formulation, had a significant effect on the cell viability. The effect of the sample concentration on the viability can be clearly seen from the scatter plot (Appendix A), whereby increasing the concentration, the score of the results decreased in the direction of the first dominant component of the model.

### 3.6. Cell Cytotoxicity Assay (LDH Assay)

The results from the cytotoxicity assay on the hCMEC/D3 cells treated with three different concentrations of the prepared lipid nano-systems after 4, 24, and 48 h are shown in Figure 5. From the graphs, it can be observed that the carriers did not show a high % of cytotoxicity at all the time points examined, indicating that fusion and the uptake of the nano-systems by the human cerebrovascular endothelial cells did not lead to membrane disruption [43,44].

In addition, tests based on MTS and adenosine triphosphate (ATP) have been known to measure the intensity of cellular metabolism as an indirect parameter of cell viability, but do not provide information on whether cell death has occurred [45,46]. As previously mentioned, the low percentage of cytotoxicity indicated that there was no disruption of the cell membrane and, accordingly, that no cell necrosis occurred. The discrepancy among the results obtained from the MTS and LDH assays for the highest concentrations of the NLCPEG and NLCPOL samples could be attributed to the consumption of cellular energy, resulting in a reduction in the metabolic activity of cells, which is a key parameter for determining the cell viability in an MTS assay.

In addition to the previously mentioned, the results of the cytotoxicity experiments on the four formulations were analyzed using a PLS model with two main components. According to the VIP plot, the dominant factor influencing the cytotoxicity was the time of exposure, as well as the type of formulation (Appendix A). From the scatter plot, distinctive behavior of the samples can be seen in relation to the incubation time, where both components equally contribute to explaining this effect (Appendix A). The influence of the formulation on the cytotoxicity is also evident, where it can be noticed that the data from the liposomal formulation with PEG were located in a separate cluster and are characterized by a higher cytotoxicity than that of the other formulations (Appendix A).

### 3.7. Internalization Studies

As previously mentioned, NLs and NLCs have been reported to successfully deliver drugs across the BBB [47]. In general, the carrier composition and its physical–chemical properties, such as size, shape, surface charge, surface hydrophobicity/hydrophilicity, and surface functionalization, control the mechanism and level of cellular uptake. Upon cellular uptake, the next crucial step is the intracellular trafficking of nanocarriers, which has impact on their final colocalization within the cellular compartments, their cytotoxicity, and their therapeutic efficacy [17]. With the aim of observing the internalization by the BBB cells and the subsequent intracellular trafficking, the cellular uptake of the NL and NLC by the hCMEC/D3 and SH-SY5Y cell lines was investigated using Fluorescent live-cell imaging and Confocal laser scanning Microscopy imaging.

The images obtained using fluorescent microscopy on the living SH-SY5Y cells showed that, after 1 h of incubation time, the internalization of the prepared nano-formulations was not completed yet (Figure 6). However, it was clearly visible that the process of cellular uptake had started and, at this phase, they had been concentrated and accumulated around the cell membrane. 

In Figure 7, the images obtained of the prepared nano-formulations after 4 h of incubation time with the living SH-SY5Y cells are presented. It could be observed that all four formulations were successfully internalized in the cells within 4 h. In addition, the colocalization of the nanoliposomes and nanostructured lipid carriers could be noticed and confirmed in the living SH-SY5Y cells, where the nanoparticles are marked with red fluorescence and the endosomes with green. Additionally, in regard to the NLC formulations, they could also follow the transcytosis pathway of internalization, where nonspecific passive diffusion was the predominant mechanism. This fact could have been due to the hydrophobic nature of the lipid carriers, making them suitable candidates for facilitated transport through apical cell membranes [48].

A 3D reconstruction of the images obtained using confocal microscopy showed that both the NLs and NLCs successfully penetrated across the cell membrane in order to be internalized in the hCMEC/D3 (Figure 8) and SH-SY5Y cell lines (Figure 9).

Namely, a significant fraction of nanoliposomes (NLPEG and NLPOL) was accumulated around the perinuclear zone with a reduced cytoplasmatic distribution (Figure 8a,b). This phenomenon probably occurred due to the ability of NLs to penetrate across the BBB by avoiding the endosomal/lysosomal pathway, subsequently finding a way to interact with the complex nuclear membrane [49,50]. On the other hand, Figure 8c,d reveal that the nanostructured lipid carriers were dispersed in the cytoplasm of the cells, probably concentrated in the intracellular vesicles (endosomes or lysosomes), which could have been related to their endocytic mechanism of internalization [51]. Taking into consideration that the mean size of the prepared NLC was around 100 nm, the obtained results were in accordance with the literature data, suggesting that endocytosis is a size-dependent process and that nanoparticles with a median diameter around 100 nm or smaller could be endocytosed more easily by cells [52,53,54].

According to the literature data, the steric stabilization of nano-systems results in successful internalization by the cells of the BBB, as well as neurons. In regard to the stabilization of the nano-systems with PEG, despite reduced opsonization and a prolonged circulation time, the increased surface charge of the particles affected their internalization in the negatively charged cells [21,55]. In addition, Shubra et al. [56] reported that Poloxamers have the ability to be incorporated into cell membranes and modify their microviscosity, as well as to decrease ATP (Adenosine triphosphate) levels, thus depleting the activity of efflux transporters, which could result in enhanced BBB penetration. This is possibly related to the surface activation of the cell membrane by the amphiphilic Poloxamer 407 [57]. So far, no similar experimental results related to this phenomenon have been reported. Therefore, further quantitative investigation studies are needed to be carried out in order to reveal the nature of this phenomenon.

### 3.8. Cellular Uptake Studies

The cellular uptake of the NL and NLC was studied on the hCMEC/D3 and SH-SY5Y cell lines, as they are some of the most well-characterized cell lines that have been demonstrated to imitate most of the characteristics of the BBB, as well as its neuronal function, respectively [58,59]. The results from the uptake studies on the hCMEC/D3 cells showed a significant difference in the amount of internalized NL (>0.2 µg) compared to NLC (~0.17 µg) (Figure 10a) at 37 °C. No significant difference was observed in the cellular uptake after the incubation with NLPEG at 4 °C on the hCMEC/D3 cell line, in contrast to NLPOL, where the relative percentage of uptake was 71.05%, expressed as the relative percentage of internalized particles at 37 °C. This obtained difference was probably due to the polymer for stealth stabilization. These results are in accordance with those of previous investigations, suggesting that, most likely, poloxamer contributes to cellular uptake via endocytosis, rather than passive diffusion [60]. In regard to the NLC, it could be noticed that the amount of cellular uptake decreased when both NLCPEG and NLCPOL were incubated at 4 °C, with percentages of 70.39% and 76.99%, respectively, which probably refers to the fact that the cellular uptake of NLC follows an ATP-dependent internalization mechanism, which is the main reason for consuming ATP and reducing energy metabolism, as seen in the previously presented results from the MTS cell viability assay.

In order to gain insight into the exact mechanism of cellular internalization, uptake studies using inhibitors for clathrin- and caveolae-mediated endocytosis were performed. It could be noticed that there was no significant decrease in cellular uptake when the NPs were incubated with chlorpromazine and indomethacin, except for NLPOL (Figure 10a). 

When interpreting the results from the cellular uptake experiments using endocytic pathway inhibitors, it should also be taken into consideration that blocking one uptake pathway can result in the activation of other endocytic mechanisms, which again confounds the data [61]. More precisely, it is important to mention that nanomedicines can be processed via different endocytic pathways (both, productive and nonproductive), and therefore, the identification and study of the exact mechanism of cell internalization become a major technical challenge [62].

The PLS model of data from the uptake studies of the different lipid formulations on both cell lines showed that the cell type was a significant independent variable that affected the particle uptake (Appendix A); therefore, it was decided to apply a discriminant PLS for each cell line. The difference in the nano-formulations’ uptake between the hCMEC/D3 and SH-SY5Y cell lines was probably due to cell-specific preferences, taking into consideration the different anatomical and physiological features of both cell lines, which further affect their mechanism and amount of internalization. Namely, the hCMEC/D3 cell line has already been successfully used as a BBB model in many studies, further attesting to its high quality and potential to replace primary cells for in vitro BBB studies. These brain endothelial cell lines retain the critical features of primary cells, such as their expressions of endothelial cell markers, transporters, and tight junctional proteins [63], which strongly affect the transport and uptake of different types of nanoparticles. On the other hand, in their undifferentiated form, SH-SY5Y cells are morphologically characterized by neuroblast-like, non-polarized cell bodies with few truncated processes. Undifferentiated SH-SY5Y cells continuously proliferate, express immature neuronal markers, and lack mature neuronal markers; thus, this type of cells is considered to be most reminiscent of immature, positive cells for tyrosine hydroxylase and dopamine-β-hydroxylase characteristic catecholaminergic neurons [64].

The data analysis of the hCMEC/D3 cell line indicated that the type of formulation had a dominant influence (Appendix A). During the analysis of the scatter plot (Appendix A), it could be seen that the type of formulation had a dominant influence, while the surface polymer (POL or PEG) had no significant influence on the cellular uptake. From Appendix A, it could be seen that energy deprivation significantly affected the uptake of all the formulations, while the inhibitors for endocytosis had no significant effect.

The SH-SY5Y cell line demonstrated substantially higher permeability values when the formulations were incubated at 37 °C, in contrast to the particles incubated with the hCMEC/D3 cell line (Figure 10b). Since neural membranes contain high levels of cholesterol [65], the higher amount of cellular uptake for the NLs compared to the NLCs was probably related to the presence of cholesterol in the nanoliposomal formulations. This is also in accordance with the results obtained by Lee et al. [66], who demonstrated that SH-SY5Y neurons show a preference for cholesterol containing liposomes. From Figure 10b, it could also be noticed that the percentage of cell internalization after 2 h of incubation of the formulations at 4 °C significantly decreased (21.45%, 16.53%, 24.09%, and 23.24% for NLPEG, NLPOL, NLCPEG, and NLCPOL, respectively), which indicates that these nano delivery systems followed an ATP-dependent mechanism of cellular internalization.

The model describing the SH-SY5Y cell experiments showed that the conditions of the experiment had the greatest influence on the amount of particles taken, with energy deprivation showing the largest VIP factor (Appendix A). On the scatter plot, separate groups according to the conditions of the experiment could be depicted, whereby the experiments under energy deprivation were given the lowest scores. Additionally, the effect of the inhibitors for endocytosis were noticed to be less significant. Regarding the influence of the type of formulation, it could be concluded that the experiments on this cell line showed that the formulations did not have distinctively different behavior in terms of cellular uptake, which could be seen from both the VIP plot (Appendix A) and the scatter plot (Appendix A).

From all the obtained results, it could be seen that the composition of the nano-systems, as well as the surface modification with PEG or POL, resulted in a formulation solution that was characterized by an optimal size and a low percentage of adsorbed plasma proteins, all of which were in favor of a prolonged in vivo circulation time, as well as efficient and safe transport across the BBB, targeted delivery, and internalization in neurons, respectively. The overall results for the potential of the investigated lipid nano-systems indicate the possibility of their further in vivo evaluation in order to develop a final dosage form with practical, efficient, and safe application in the treatment of brain diseases.

## 4. Conclusions

Based on the obtained TEM micrographs, it could be concluded that all the formulations were characterized by a spherical form and smooth surface, the z-average diameter was 105–120 nm, following a unimodal particle size distribution and a negative zeta potential (~−30 mV). The stability studies showed that, when incubated in a physiologically relevant medium, the nano-systems did not change their mean size significantly. On the other hand, there was a slight increase in the particle mean size of the prepared formulations after 24 h of incubation time in human plasma, which was probably due to the process of protein corona formation. The ATR-FTIR spectra confirmed the presence of stealth polymers on the nano-carriers’ surfaces. Additionally, the presence of PEG and POL contributed to a lower plasma protein adsorption when compared to the formulations with no stealth polymer. From the obtained results, it could be concluded that the prepared lipid nano-systems in the tested concentrations did not show cytotoxicity, with the exception of the reduced percentage of cell viability with the highest concentration of NLCPEG and NLCPOL, which was probably due to the consumption of cellular energy and the decrease in the metabolic activity of the cells. The successful internalization of the lipid nano-systems in the hCMEC/D3 and SH-SY5Y cell lines supports their potential for safe targeting and transport through the BBB, as well as the effective treatment of CNS diseases. However, in regard to the evaluation of the exact uptake mechanism of the nanoparticles by the abovementioned cell culture lines, it should be highlighted that nano-carriers can be transported and internalized via multiple pathways, and therefore additional studies should be conducted.

## Figures and Tables

**Figure 1 pharmaceutics-15-02082-f001:**
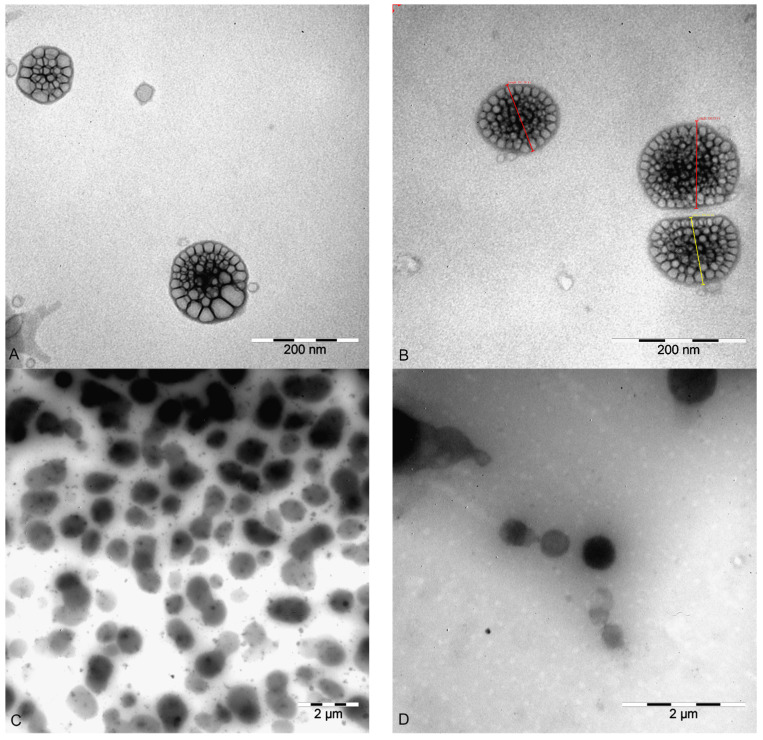
TEM images of the prepared nano-systems: (**A**) NLPEG; (**B**) NLPOL; (**C**) NLCPEG; and (**D**) NLCPOL.

**Figure 2 pharmaceutics-15-02082-f002:**
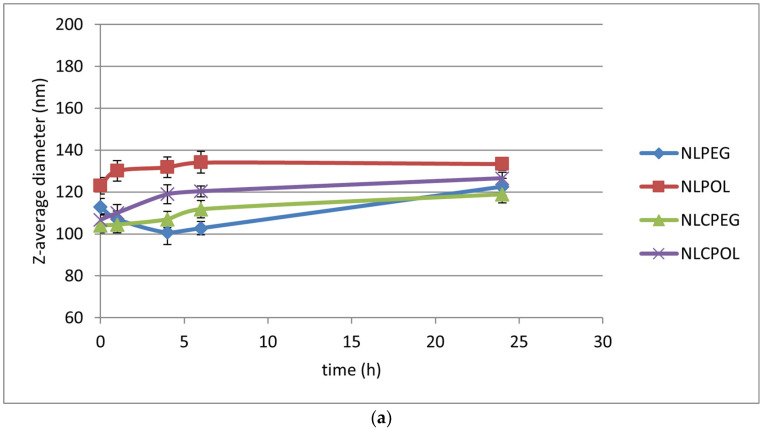
Mean particle size of NL and NLC samples incubated in a period of 24 h in: (**a**) phosphate buffer pH 7.4; and (**b**) human plasma.

**Figure 3 pharmaceutics-15-02082-f003:**
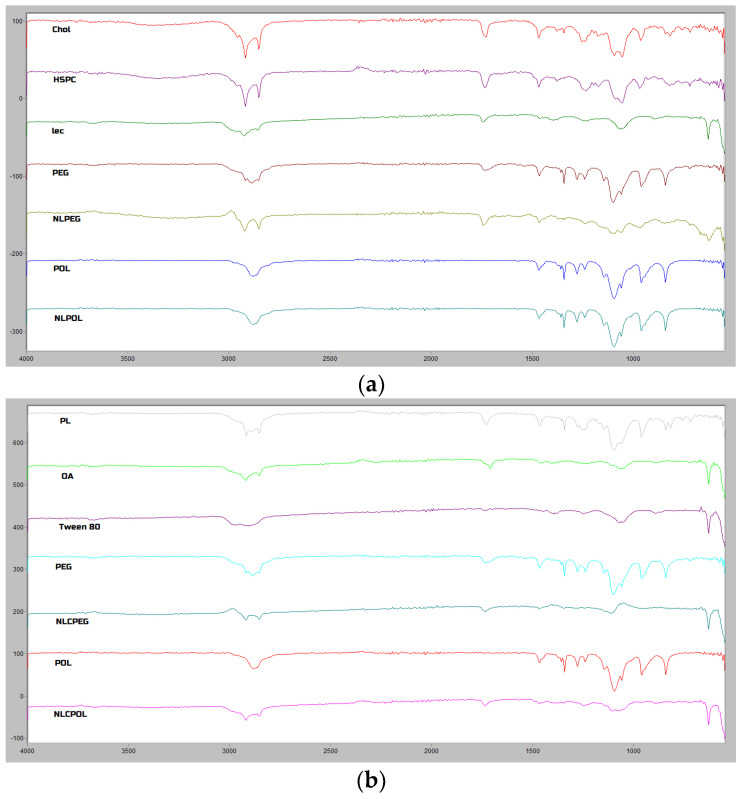
ATR-FTIR spectra of formulation components and; (**a**) NLPEG and NLPOL; and (**b**) NLPEG and NLPOL.

**Figure 4 pharmaceutics-15-02082-f004:**
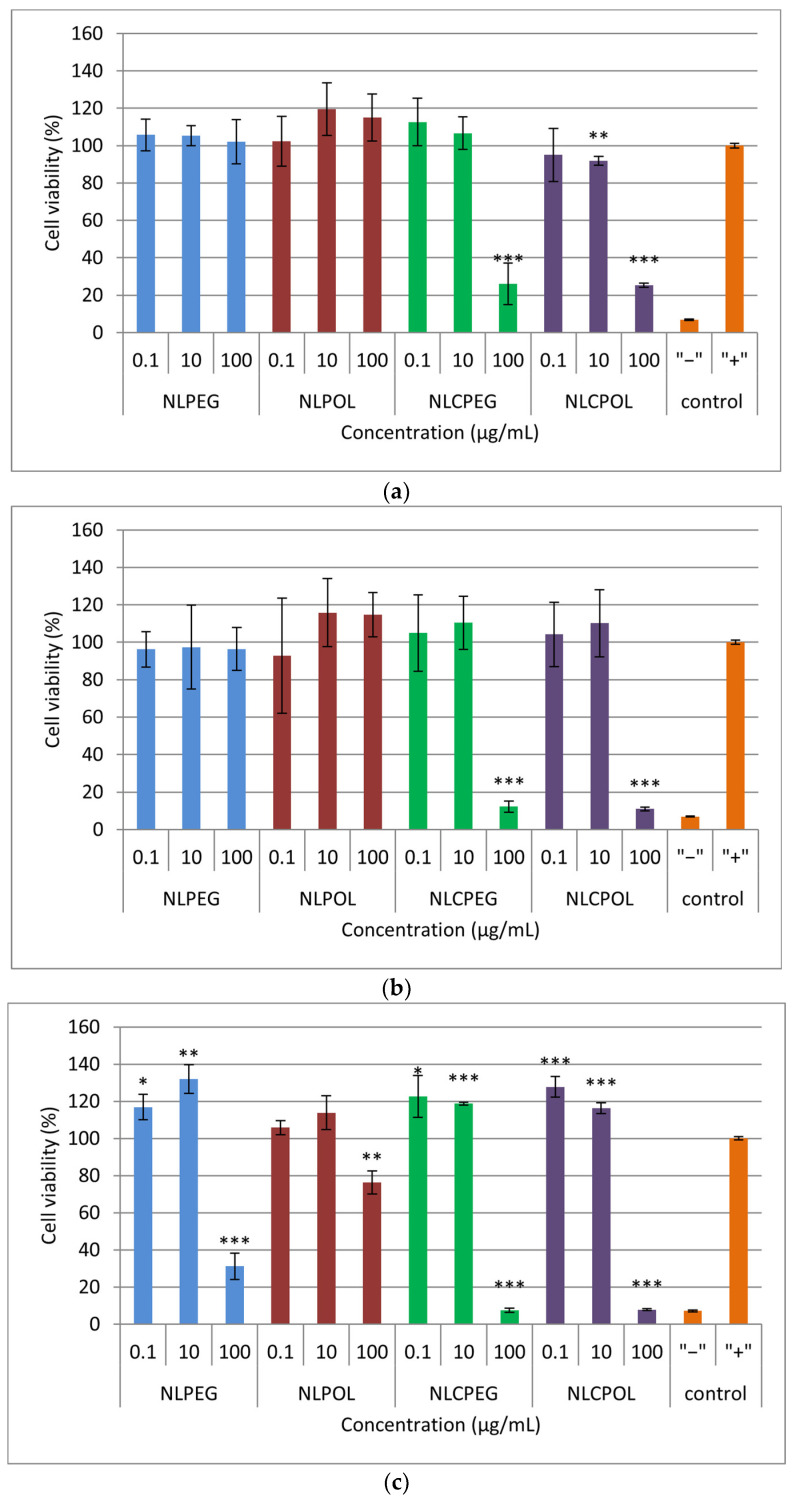
Cell viability after: (**a**) 4 h; (**b**) 24 h; and (**c**) 48 h incubation with the prepared samples. Repeated measures were statistically tested by one way ANOVA, where *** *p* < 0.001, ** *p* < 0.01, and * *p* < 0.05, versus the positive control. NLPEG has been marked with blue color, NLPOL with red, NLCPEG with green, NLCPOL with purple and the controls with orange.

**Figure 5 pharmaceutics-15-02082-f005:**
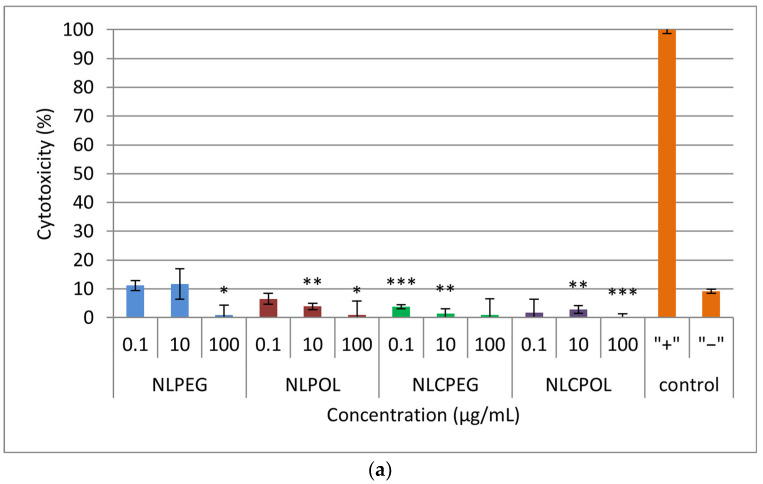
Cell cytotoxicity after: (**a**) 4 h; (**b**) 24 h; and (**c**) 48 h incubation with the prepared samples. Repeated measures were statistically tested by one way ANOVA, where *** *p* < 0.001, ** *p* < 0.01, and * *p* < 0.05, versus the negative control. NLPEG has been marked with blue color, NLPOL with red, NLCPEG with green, NLCPOL with purple and the controls with orange.

**Figure 6 pharmaceutics-15-02082-f006:**
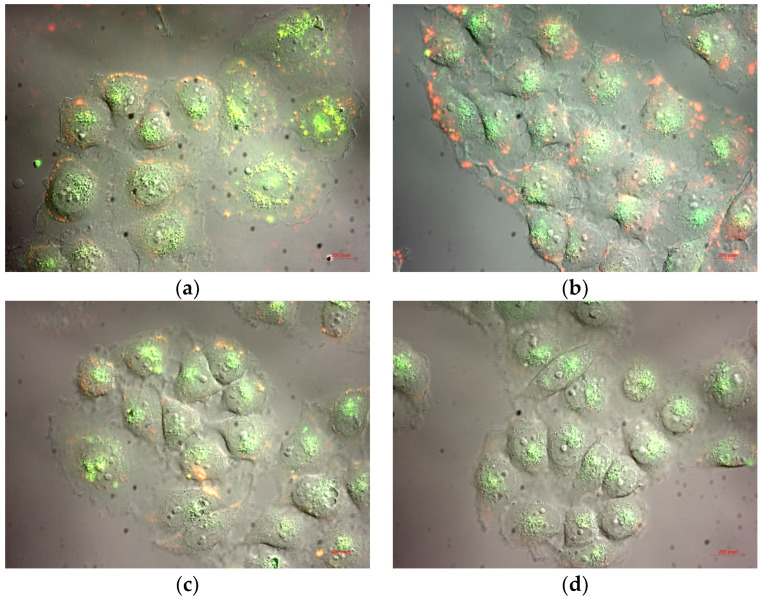
Internalization in SH-SY5Y after 1 h incubation time with: (**a**) NLPEG; (**b**) NLPOL; (**c**) NLCPEG; and (**d**) NLCPOL.

**Figure 7 pharmaceutics-15-02082-f007:**
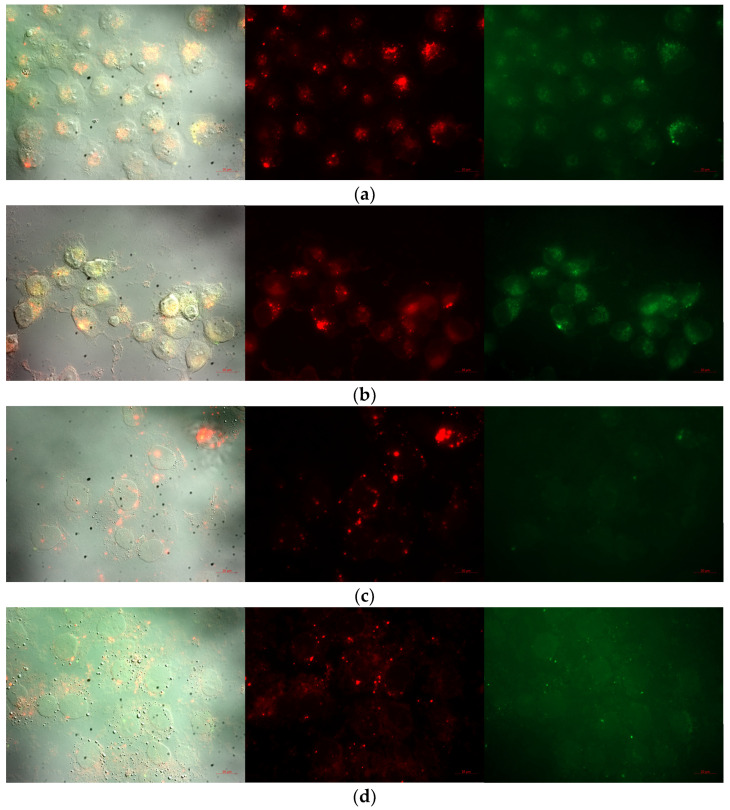
Incubation in live SH-SY5Y after 4 h, where the samples were marked with red fluorescence, the nucleus with blue, and the endosomes with green fluorescence: (**a**) NLPEG; (**b**) NLPOL; (**c**) NLCPEG; and (**d**) NLCPOL.

**Figure 8 pharmaceutics-15-02082-f008:**
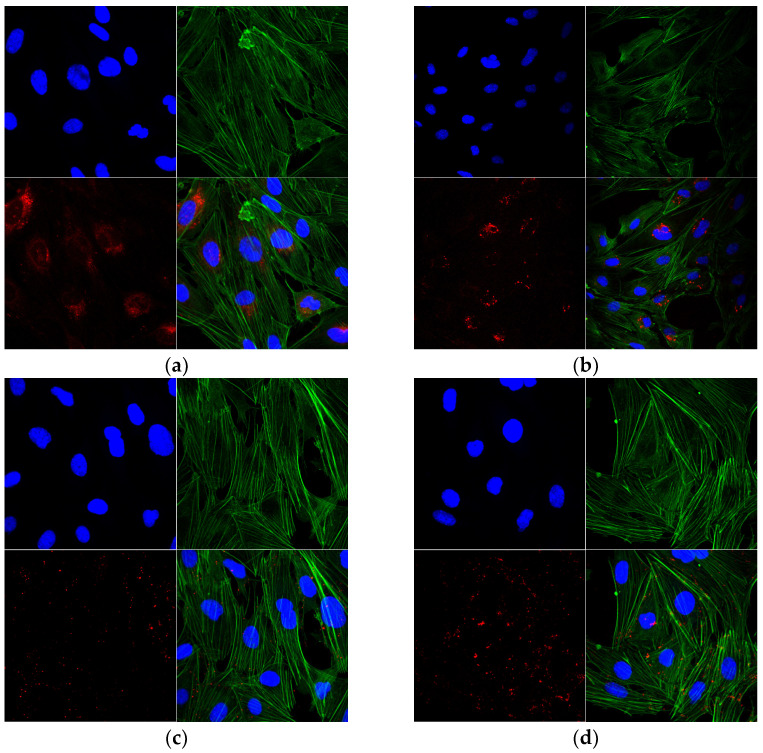
Confocal microscopy of internalization and distribution of prepared samples in hCMEC/D3 cell line: (**a**) NLPEG; (**b**) NLPOL; (**c**) NLCPEG; and (**d**) NLCPOL, where the cytoskeleton is marked with green dye, the nucleus with blue, and the nano-systems with red dye, respectively.

**Figure 9 pharmaceutics-15-02082-f009:**
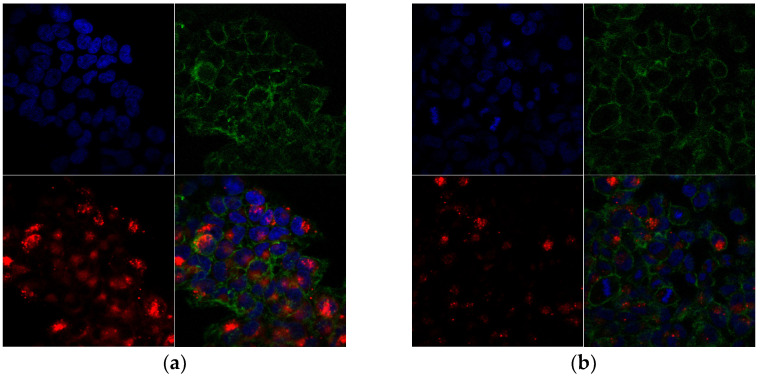
Confocal microscopy of internalization and distribution of prepared samples in SH-SY5Y cell line: (**a**) NLPEG; (**b**) NLPOL; (**c**) NLCPEG; and (**d**) NLCPOL, where the cytoskeleton is marked with green dye, the nucleus with blue, and the nano-systems with red dye, respectively.

**Figure 10 pharmaceutics-15-02082-f010:**
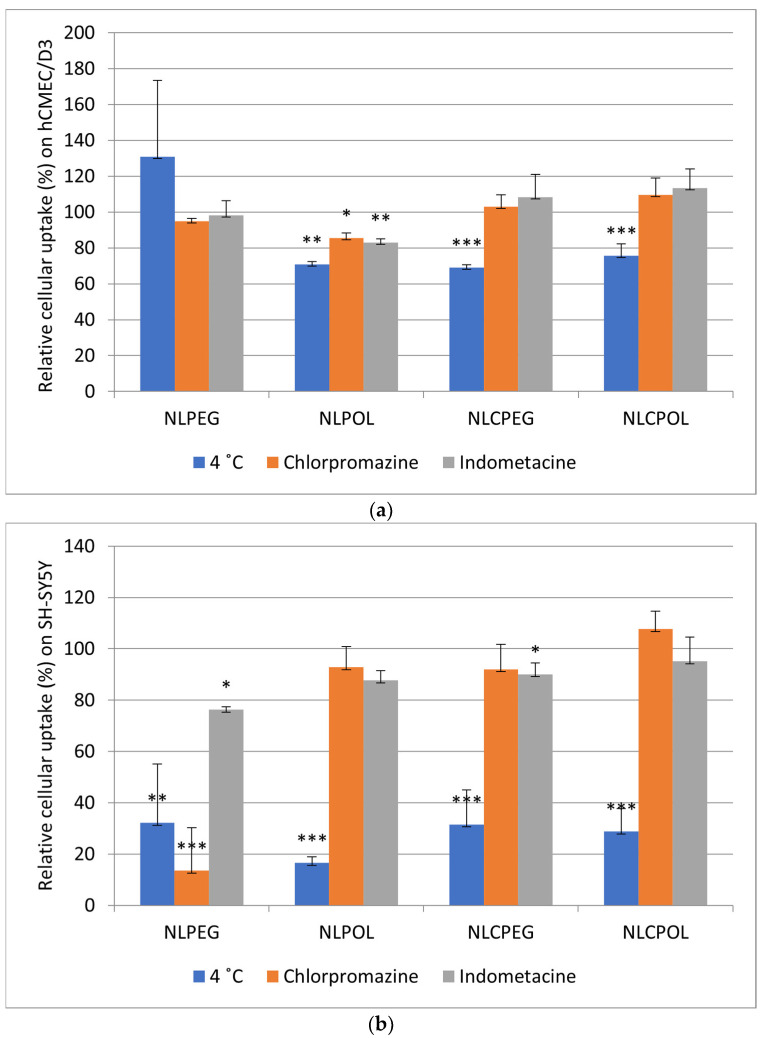
Amount (µg) of internalized particles of NL and NLC after 2 h incubation time in (**a**) hCMEC/D3 and (**b**) SH-SY5Y cell line; at 4 °C for inhibition of all energy-dependent mechanisms; with chlorpromazine (CPZ) for inhibiting clathrin-mediated endocytosis and with indomethacin (INDO) for inhibiting caveolae-mediated endocytosis, where *** *p* < 0.001, ** *p* < 0.01, and * *p* < 0.05, versus the percent of internalized particles at 37 °C.

**Table 1 pharmaceutics-15-02082-t001:** Physical–chemical properties of prepared nano-systems.

Sample	Z-Average Diameter ± SD (nm)	PDI ± SD	ZP ± SD (mV)
NLPEG	112.900 ± 4.726	0.164 ± 0.035	−26.500 ± 2.928
NLPOL	123.000 ± 1.365	0.270 ± 0.006	−31.201 ± 3.326
NLCPEG	104.000 ± 3.200	0.191 ± 0.065	−24.600 ± 0.833
NLCPOL	106.700 ± 2.612	0.181 ± 0.024	−24.301 ± 0.850

**Table 2 pharmaceutics-15-02082-t002:** Adsorbed protein (%) onto the NL and NLC surfaces.

NL			NLC		
0	PEG	POL	0	PEG	POL
29.254 ± 0.534%	8.042 ± 4.032%	14.802 ± 6.609%	22.727 ± 5.221%	13.054 ± 2.577%	13.636 ± 2.293%

## Data Availability

The data presented in this study are available on request from the first or corresponding author.

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
