# Peer review of "Comparative Studies of the Uptake and Internalization Pathways of Different Lipid Nano-Systems Intended for Brain Delivery"

_pharmaceutics, 2023, doi:10.3390/pharmaceutics15082082_

Round 1

Reviewer 1 Report

The papers deals with the preparation of Lipid nano-systems and their characterization also with a series of well-established in vitro tests that could assess the interactions with hCMEC/D3 and SH-SY5Y cell lines as a model for blood-brain barrier and neuronal function.

Additionally, the paper confirms that the presence of the PEG and POL contributed to  lower plasma protein adsorption. The prepared lipid nanosystems do not show cytotoxicity.

The paper is well structured and the conclusions are well documented by experimental findings, thus the paper can be of interest to deepen the important aspect of internalization pathways definition.

The paper need some minor revision:

Line 47 “BBB is considered the most challenging” obstacle?

Line 101: physical-chemical instead of physic-chemical

Line 102-104 I do not agree at all with the following statement, which, moreover, is not true as demonstrated by the high number of the published work from al that correlates the physicochemical characteristics of nanocarriers with their in vivo activities

Line 148 Add phosphate buffer pH 7.4 concentration

Line 179: please specify if the contact was 100% human plasma or if other percentages.

TEM micrograph: please specify which type of structure is in images A and B. Moreover dimensions revealed by TEM are not in agree with those obtained by DLS.

Line 518- Results of internalization studies and of cellular uptake studies must be better explained in a more precise way. Sometimes it is not clear what authors explain and what they want to highlight.

English language must be revised throughout the text

Author Response

see attachment for reviewer 1

Reviewer 2 Report

The manuscript by Andreas Zimmer et al. was interesting, and fell within the scope of Pharmaceutics. Brain delivery was a current research hotspot in the field of pharmaceutical sciences, and thus this work might provoke a certain impact. However, several points were still pending addressed before final acceptance, and a Minor Revision was suggested.

Detailed comments:

Q1: The first paragraph of Introduction should be accompanied with references.

Q2: It was suggested to prepare an illustration for the lipid carriers’ preparation procedures.

Q3: According to the Materials and Methods Section, the procedures might adhere to EP 10.0. Please notice that EP10.0 was not the latest version. It would be better to consult the newest version.

Q4: The significant digits of data in Table 1 should be uniformed.

Q5: For the protein adsorption results, no S.D. values were shown. Please supplement the S.D.

Q6: The micrographs of cellular tests (Figure 6~9) should be accompanied with scale bars.

Q7: Table 3 could be transformed to a Figure to gain more visibility.

Q8: Before the Conclusion Section, the authors could discuss the clinical and industrial translation aspects of the lipid carriers.

Author Response

see attachment for reviewer 2

Reviewer 3 Report

The submitted manuscript “Comparative studies of the uptake and internalization pathways of different lipid nano-systems intended for brain delivery”. The topic of this manuscript is interesting for readers of Pharmaceutics. I have however some reservations and I therefore recommend publication of this manuscript only if the authors can address the major issues noted below.

1.       Line 24- increasement, it should be increase in….

2.       The authors may want to clearly state what are key research gaps here which are not achieved in literature and what is novelty and significance of their work in the introduction.

  1. The aim of the study is poorly described in the last paragraph of the introduction. Please, improve it

4.       Line 38- suitable references should be incorporated.

5.       Figure 3, is not clear, it should improve.

6.       Figure 4 &5 , statistics asterics should be flagged.

7.       If possible, please provide at least 3 months accelerated stability data to understand the stability of developed in terms of size, PDI, and ZP.

minor english correction required.

Author Response

see attachment for reviewer 3

Round 2

Reviewer 3 Report

Accept in present form.